# CO_2_ Laser for Esthetic Healing of Injuries and Surgical Wounds with Small Parenchymal Defects in Oral Soft Tissues

**DOI:** 10.3390/diseases11040172

**Published:** 2023-11-28

**Authors:** Yuki Daigo, Erina Daigo, Hiroshi Fukuoka, Nobuko Fukuoka, Jun Idogaki, Yusuke Taniguchi, Takashi Tsutsumi, Masatsugu Ishikawa, Kazuya Takahashi

**Affiliations:** 1Department of Geriatric Dentistry, Osaka Dental University, 2-2-14 Higashitanabe, Higashisumiyoshi-ku, Osaka 546-0032, Japan; comcomjun@hotmail.com (J.I.); kazuya-t@cc.osaka-dent.ac.jp (K.T.); 2Nogami Dental Office, 4-22-18 Nishiimagawa, Higashisumiyoshi-ku, Osaka 546-0042, Japan; daigou_e0120@yahoo.co.jp; 3Fukuoka Dental Office, 704-4 Torai, Satsuma-gun, Kagoshima 895-1811, Japan; hn-fukuoka@po4.synapse.ne.jp (H.F.); nobuko-aoki@hotmail.co.jp (N.F.); 4Section of Oral Implantology, Department of Oral Rehabilitation, Fukuoka Dental College, 2-15-1 Tamura, Sawara-ku, Fukuoka 814-0193, Japan; yuusuke@fdcnet.ac.jp; 5The Center for Visiting Dental Service, Department of General Dentistry, Fukuoka Dental College, 2-15-1 Tamura, Sawara-ku, Fukuoka 814-0193, Japan; kanade09@fdcnet.ac.jp; 6Bees Dental Office, 6-904 Befudanchi, Jyonan-ku, Fukuoka 814-0106, Japan; beezaemonn@yahoo.co.jp

**Keywords:** CO_2_ laser, high-intensity laser therapy, photobiomodulation therapy, scar, mucosal epithelium, open lip, socket preservation, free gingival graft

## Abstract

A number of studies have recently demonstrated the effectiveness of CO_2_ laser irradiation for the repair and regeneration of scar tissue from injuries or surgical wounds. However, such studies of the oral mucosa are highly limited. Previous studies using CO_2_ laser irradiation have indicated that two factors contribute to esthetic healing, namely, artificial scabs, which are a coagulated and carbonized blood layer formed on the wound surface, and photobiomodulation therapy (PBMT) for suppressing wound scarring and promoting wound healing. This review outlines basic research and clinical studies of esthetic healing with the use of a CO_2_ laser for both artificial scab formation by high-intensity laser therapy and PBMT in the treatment of injuries and surgical wounds with small parenchymal defects in oral soft tissues. The results showed that the wound surface was covered by an artificial scab, enabling the accumulation of blood and the perfusion necessary for tissue regeneration and repair. Subsequent PBMT also downregulated the expression of transformation growth factor-b1, which is involved in tissue scarring, and decreased the appearance of myofibroblasts. Taken together, artificial scabs and PBMT using CO_2_ lasers contribute to the suppression of scarring in the tissue repair process, leading to favorable esthetic and functional outcomes of wound healing.

## 1. Introduction

During the healing of injuries and surgical wounds with small parenchymal defects in oral soft tissues, scarring (e.g., formation of hypertrophic scars, contracture and epithelial concavity) sometimes occurs, affecting esthetic and functional outcomes. Scarring and associated problems can also occur due to suturing performed to achieve early wound closure and prevent wound infections [1,2] and due to the use of a wound dressing [3,4].

Recently, the effectiveness of CO_2_ lasers (a fractional CO_2_ laser, in particular) for the treatment of epithelial scars from trauma, surgical wounds and acne was reported in clinical studies using biopsy samples [5,6,7,8,9], in studies using a scar scale or a visual analog scale [10,11,12,13,14,15,16] and in basic studies using experimental animals [17,18,19,20,21,22,23,24].

We have been studying CO_2_ laser treatment of tooth extraction sockets and reported basic research results such as the promotion of wound healing, preservation of the alveolar crest and suppression of mucosal scarring around the extraction sockets [25,26,27,28]. We also reported clinical research results on CO_2_ laser treatment that suppressed scarring during the healing of open lip vermillion wounds caused by injuries, resulting in esthetic tissue regeneration [29]. Two factors contribute to the esthetic wound healing achieved by CO_2_ laser treatment. One is the presence of “artificial scabs” formed by coagulation and carbonization of the surface of blood accumulated in injuries and surgical wounds by high-intensity laser therapy (HILT). The other is photobiomodulation therapy (PBMT), which is expected to promote wound healing and suppress scar formation.

This review outlines basic research and clinical studies on esthetic healing by use of a CO_2_ laser for both artificial scab formation by HILT and PBMT in the treatment of injuries and surgical wounds with small parenchymal defects in oral soft tissues and provides key points of treatment procedures based on the findings from some clinical case examples.

First, we briefly explain scar formation in primary healing and secondary healing.

## 2. Wound Healing and Scarring

The presence of myofibroblasts is essential for the closure of injuries and surgical wounds. Myofibroblasts, which have smooth muscle cell-like contractile activity, are differentiated from fibroblasts in granulation tissue upon expression of TGF-b1. Then, scarring and scar contracture occur, and early wound closure is promoted by total coverage of wounds by epithelia [30,31,32,33].

However, this healing mechanism can cause unfavorable esthetic and functional outcomes because of the development of hypertrophic scars, scar contracture and epithelial concavity on the wound surface [34,35,36]. The degree of scar formation varies depending on the size, area and depth of parenchymal defects.

### 2.1. Primary Healing and Scar Formation

Primary wound healing occurs when an incision made by a scalpel is closed by tight suturing of the dermal edges. In principle, primary healing can be achieved when wounds are not contaminated and do not require debridement. However, it is possible after thorough debridement if the wound is closed by suturing the dermal edges without leaving dead space. Also, when suturing, it is important to carefully align the epithelium on one dermal edge with that of the other and the submucosa on one dermal edge with that of the other.

The following problems are associated with primary healing: (1) a linear scar forms at the surface plane of the closed wound edges because of the scar tissue involved and contraction of the wound surface; (2) when suturing or reefing of the muscle layers is performed, the wound surface and surrounding tissue are pulled, leading to motor dysfunction due to contracture and tension in the future; and (3) when suturing or reefing of the wound edges is not tight enough, dead space forms subcutaneously or at the deep position beneath the wound, and then granulation tissue forms to fill the dead space, which is eventually replaced by scar tissue.

### 2.2. Secondary Healing and Scar Formation

Secondary healing occurs in open wounds that lack epithelia. After thorough debridement, blood on the wound surface coagulates, and defects are filled by the granulation tissue formed via the deposition of fibroblasts and angiogenesis. Then, fibroblasts differentiate into myofibroblasts in granulation tissue, and the granulation tissue is replaced by scar tissue, resulting in repair and regeneration. At the same time, the epithelium around the wound extends to cover the entire area of the wound, leading to healing.

Recently, to prevent scar formation like that described above, dressing therapy, where the wound is covered by a dressing, has been used for open wounds [3,4]. This facilitates early wound healing by keeping the wound surface adequately moist, or more precisely, by keeping blood and effusion containing cytokines that induce migration, division and proliferation of cells within the wound, and by protecting regenerated immature epithelia and granulation tissue from contamination and infection (Figure 1a). However, the following problems are associated with this treatment: (1) there is a risk of aggravation of infection in the closed environment (main causes are the presence of bacteria, inadequate debridement, oxygen deficiency); (2) epithelial healing may be inadequate due to the accumulation of excessive effusion in the highly closed conditions; (3) there is a risk of damage to the wound surface upon removal of the wound dressing; and (4) fixation of a wound dressing over highly mobile tissue is difficult, causing misalignment or opening of wound edges, which may result in dead space formation and consequent formation of a large amount of scar tissue.

Next, we take into account the characteristics of CO_2_ lasers and explain to what extent the CO_2_ laser treatment can suppress scar formation in primary healing and secondary healing of injuries and surgical wounds with small parenchymal defects in oral soft tissues.

## 3. Characteristics of CO_2_ Laser

Dental CO_2_ lasers emit long-wavelength light (10.6 mm) that is strongly absorbed by water. Therefore, upon irradiation of the skin, the thermal energy of a CO_2_ laser is mostly absorbed before reaching a depth of 0.05 mm beneath the skin surface, causing no impact on the deep tissue [37,38,39]. Similarly, in the irradiation of blood, coagulation and carbonization occur only at the surface, and blood under the surface layer is unaffected.

However, excessive CO_2_ laser irradiation due to inappropriate conditions (e.g., output, irradiation mode, irradiation time, irradiation distance) and inappropriate maneuvers can cause irreversible changes and delayed healing.

## 4. Secondary Healing-like Effect Using a CO_2_ Laser

The two factors of artificial scabs and PBMT are involved in achieving secondary healing using a CO_2_ laser. We explain these factors below.

### 4.1. Presence of Artificial Scabs

Artificial scabs, formed by coagulation and carbonization of blood at the wound surface by HILT, are necessary for the repair and regeneration of parenchymal defects associated with injuries and surgical wounds to obtain the original tissue form [25,26,27,28,29]. These artificial scabs play a role similar to that of wound dressings in dressing therapy (Figure 1b). They have a “space-making effect” to retain effusion and blood required for repair and regeneration of tissue to its original form, preserve the moist condition for moist wound healing and protect immature epithelial and granulation tissues from contamination and infection, thereby facilitating early wound healing. Problems associated with wound dressings, explained in Section 2.2, are absent in the case of artificial scabs: problems (1) and (2) are unlikely because artificial scabs are breathable; problem (3) is unlikely because they fall off naturally rather than requiring forced removal; and problem (4) is unlikely because laser soldering of artificial scabs formed over the mobile tissue to the surrounding tissues reduces the risk of peeling [40,41].

### 4.2. Preventive Effect of PBMT on Scarring

PBMT is expected to promote wound healing and prevent scar formation. This review focuses on scar prevention by PBMT in the healing of oral soft tissue. Those who are interested in the promotion of healing by PBMT are encouraged to refer to our previous studies [25,26,27]. PBMT inhibits differentiation of fibroblasts to myofibroblasts in granulation tissue and inhibits or improves scarring.

Recently, the effectiveness of PBMT using a CO_2_ laser for inhibiting or improving scar formation associated with trauma, burns, surgical wounds and acne in skin tissue was demonstrated in clinical studies using a biopsy or a scar scale. Pathohistological and biochemical examination of wound biopsy samples showed inhibition of collagen fiber production [5,6,9], normalization of collagen fiber orientation [5], normal regeneration of and increases in dermal collagen with elastic fibers [8,10], thinning in the stratum corneum [5,9] and downregulation of expression of TGF-b1 involved in differentiation of fibroblasts into myofibroblasts [5,7,9]. With respect to appearance, studies of scar healing, such as those using the Vancouver Scar Scale or ultrasound measurement, have demonstrated improvements in the hardness [10,14], thickness [11,16] and flexibility and elasticity [10,14] of scars. In studies of motor function at scar sites [12], studies using the Patient Scar Assessment Questionnaire showed improvements in appearance and scar awareness [10,11,12,13,14,15,16].

Basic studies using experimental animals with artificially formed scar tissue in skin tissue have shown that scar formation can be alleviated by PBMT through downregulation of TGF-b1 expression [17,20,21], which decreases expression of the myofibroblast marker α-smooth muscle actin (α-SMA) [21,22], decreases the amount and density of excessive collagen fibers produced in granulation tissue [17,18,19,20,21], decreases the disruption of fiber orientation [19,20], decreases the scar elevation index [21], decreases the micro-vessel density [18] and thins the fiber layer [20]. Also, PBMT induces downregulation of fibroblast expression in scar tissue [18,19,21], where basic fibroblast growth factor (bFGF) plays a role. During the maturation phase of wound healing, a scar forms when many fibroblasts are present in the tissue. Secretion of bFGF induces apoptosis of fibroblasts, thereby inhibiting and reducing scar tissue formation due to the production of excessive collagen fibers. On the other hand, during the inflammatory and proliferative phases, bFGF promotes cell division to increase fibroblasts, thereby promoting repair and regeneration of the wound. During these phases, PBMT stimulates bFGF secretion to promote repair and regeneration of normal tissue without scarring [23,24].

The relationship between PBMT and TGF-b1 has been studied with respect to suppression of scarring and promotion of wound healing. Laser beams affect mitochondrial cytochrome-C oxidase, thereby influencing the production of adenosine triphosphate (ATP), which is a major source of energy for cell functions. These responses include the generation of reactive oxygen species (ROS) that activate nuclear factor-κB, which plays a role in the signaling cascade, including wound shrinkage, fibroblast differentiation and collagen production. ROS induces extracellular activation of TGF-b1. Also, downregulation of TGF-b1 signaling enhances the formation of keloids and hypertrophic scars. However, PBMT increases ATP production, but the level of ROS remains low. As a result, the downregulation of TGF-b1 by PBMT affects the reduction and inhibition of profibrotic gene synthesis and collagen synthesis. These are likely to indicate the suppression of wound tissue scarring [42,43]. On the other hand, TGF-b1 is a potent regulator of inflammatory responses and is usually upregulated in the early phase of wound healing. However, when comparing treatment with and without PBMT, we found no significant differences in TGF-b1 expression in the inflammatory phase but a significant decrease by PBMT in the proliferative phase of wound healing, indicating a suppressive effect on wound tissue scarring [26,28].

After PBMT, TGF-b2, like TGF-b1, is also involved in recruiting fibroblasts and immune cells from the circulation and wound edges to the wounded area, thereby promoting granulation tissue formation and collagen synthesis [44]. On the other hand, a different study showed the appearance of apoptotic epithelial cells and fibroblasts after PBMT [45]. This phenomenon is likely to show suppression of tissue scarring through a reduction in excessive collagen production associated with wound healing. In contrast to TGF-b1 and TGF-b2, TGF-b3 possibly reduces scarring in adults and promotes scarless healing in fetuses, but no study has provided evidence on TGF-b3 in relation to PBMT. Basic studies on PBMT using a CO_2_ laser and TGF-b are limited, and future studies are awaited.

Taken together, both clinical and basic studies have demonstrated that PBMT is effective in suppressing scar tissue formation. At present, although suppression of scar formation has been verified, complete healing is difficult solely with PBMT using a CO_2_ laser. To address this, combination with a laser of a different wavelength or an agent has been studied to improve the reliability of treatment effectiveness [7,9,17,18,19,20,21].

As described above, studies of scars in the oral mucosa, including our studies, are limited compared with those of scars in skin tissue. This can be explained by the rarity of scar formation in the oral mucosa due to differences in turnover (a few days to 2 weeks for the oral mucosa vs. approximately 4 weeks for skin tissue). However, depending on the treatment procedures used, mucosal concavity and recession occur, influencing the esthetic outcomes and motor functions of the tongue, lips and cheeks. Also, it is important to understand that the lips are complex tissues with a transition between oral mucosa and skin tissue.

Next, we describe secondary healing achieved in clinical cases by treatment procedures using a CO_2_ laser (Figure 2).

## 5. Treatment of the Lips Using a CO_2_ Laser

Achieving favorable esthetic outcomes is crucial in the facial area, including the lips, so careful postoperative treatment is important.

### 5.1. Mucocele of the Lip

A mucocele of the lip is thought to be caused by impaired outflow of mucus from a minor salivary gland due to damage to the opening of the minor salivary gland duct present under the labial mucosa.

In conventional mucocele removal, infiltration anesthesia is applied to the area around the mucocele, and an incision is made on the normal mucosa without damaging the mucocele using a scalpel. Then, the lesion is fully enucleated from the surrounding tissue using a surgical scalpel, mucosal elevator or mosquito forceps, and lastly, suturing is performed (Figure 3a–e) [46].

On the other hand, in mucocele removal using a CO_2_ laser, tissues are cut and vaporized by a CO_2_ laser instead of being cut by a scalpel, and basically, no suturing is required. The most important point of this method is to retain blood in the space generated after mucocele removal and along the lip morphology and to form an artificial scab on the surface. Then, artificial scabs are strongly soldered to the surrounding tissues to avoid detachment [40,41], and PBMT is subsequently applied several times before the end of the procedure (Figure 2a–c and Figure 3a,f–k). This method frees surgeons from suturing and also frees patients from pain associated with suturing and uncomfortable symptoms such as tension and contracture of the lip. The surgical procedure for hemangioma resection, although we have not performed it, is basically the same as that described above [47].

Recently, clinical studies have shown that this treatment procedure can reduce bleeding and pain, requires a short time and is associated with almost no or no recurrence [48,49]. However, findings regarding esthetic and motor function outcomes associated with scar formation in treated tissue have not yet been reported in detail.

Clinical studies using lasers of other wavelengths have also been reported [50,51], but no studies using biopsy have yet been reported. Also, there are specific problems associated with the use of lasers. For example, when a laser tip is in direct contact with the skin for vaporization and cutting in treatment using a laser that penetrates tissue (e.g., a diode laser), bleeding during the treatment is reduced, but heating of the surrounding tissue induces protein coagulation, which is associated with a risk of delayed wound healing and some scar formation [52,53]. Compared with CO_2_ lasers, when Er.YAG (erbium-yttrium-aluminum-garnet) lasers, which are absorbed at the surface, are used, more bleeding is expected, albeit without the effect of heating the surrounding tissue during treatment [39,52,54].

### 5.2. Open Lip Vermillion Wounds

The lips are the part of the face most prone to injury. Treatment of open lip vermillion wounds includes thorough debridement of contaminated tissue at the injured site and then advanced suturing (e.g., V-Y advancement flap [1,2]) that achieves esthetic and functional reconstruction of the lips. Although debridement is necessary, it must be kept to a minimum when treating tissues (e.g., the face) where favorable esthetic outcomes are particularly important. It is highly likely that forceful reefing results in scar formation along the wound edges that are closed and consequent interruption of the vermillion border (Figure 4a–e). Similar findings were shown in suturing in the surgery of the cleft lip [55,56]. Also, because the lips are highly mobile, wound dehiscence and the formation of dead space in the deep part beneath the wound may occur, as described in problem (4) of Section 2.2.

On the other hand, in treatment using a CO_2_ laser, debridement and artificial scab formation are accomplished. In this procedure, after minimum vaporization of contaminated tissue by the laser, blood is allowed to accumulate in the dead space formed at the site of the tissue defect and along the lip morphology, and artificial scabs are formed on the blood surface and strongly soldered to the surrounding tissues [40,41]. Then, PBMT is performed several times (Figure 2d–f and Figure 4a–c,f–k). It should be noted that the formation of artificial scabs without careful consideration results in irregular closure of the lip wound edges, potentially causing concavity, unevenness and hypertrophic scar formation. Thus, it is important to align and close the lip wound edges by light suturing and then to form artificial scabs [29].

## 6. Treatment Using a CO_2_ Laser after Tooth Extraction

The guidelines for use of dental lasers formulated by the U.S. Food and Drug Administration recommend the “coagulation of extraction sites” using a CO_2_ laser.

In the tooth extraction procedure using a CO_2_ laser, tooth extraction is performed in the conventional manner, but hemostasis is achieved by laser coagulation. Suturing or compression are mainly used for hemostasis in tooth extraction, but wound closure by suturing causes excessive tension in the surrounding mucosa and a consequent reduction in the height of the alveolar mucosa, and the gauze pieces and dental cotton rolls used for compression hemostasis absorb blood. Thus, with these conventional hemostasis procedures, an adequate amount of blood, which is necessary for alveolar bone regeneration in the extraction socket, cannot be secured in the extraction socket, resulting in vertical alveolar bone resorption and a mucosal concavity at the site of the extraction wound (Figure 5a,b and Figure 6a–e).

On the other hand, in the treatment using a CO_2_ laser, an artificial scab is formed after letting blood accumulate to the height of the surrounding mucosa of the extraction socket, and it is soldered to the surrounding mucosa [40,41], and PBMT is then performed several times (Figure 2g–i, Figure 5c and Figure 6f–j).

These scabs prevent the extension and invagination of the surrounding mucosa into the extraction socket, thereby playing a role in space-making for bone regeneration within the extraction socket. Subsequent PBMT is also important for treating extraction wounds. It suppresses the contracture of a scar developed on the mucosal epithelium through marked downregulation of TGF-b1 and α-SMA in the mucosal epithelium of extraction wounds, as shown in our previous studies [26,28] and described in Section 4.2. PBMT also plays a role in the abovementioned space-making. In rats, we also demonstrated that activation of bone remodeling and formation of new bone with a cross-linking pattern occurred in the shallow layer of an extraction socket at the depth of the CO_2_ laser light penetration. Such new bone with a cross-linking pattern serves as a bone lining under the mucosa of the extraction wound, thereby preventing extension of the mucosal epithelium into the extraction socket and making space for bone regeneration [25,26,27].

Taken together, artificial scabs and PBMT contribute to preserving alveolar bone height to the greatest extent possible and also to preventing the formation of a mucosal epithelial concavity at the site of the extraction wound.

Note that when marked alveolar bone resorption is present due to periodontal diseases or other conditions, an adequate height of the extraction socket wall, which is a part of the alveolar bone supporting a dentition, cannot be secured. Consequently, an adequate amount of blood cannot be accumulated. This results in a reduction in the regenerated bone height, causing mucosal concavity at the site of the extraction wound.

## 7. Free Gingival Grafting Using a CO_2_ Laser

In this treatment, for cases with no or little attached gingiva, a gingival flap harvested from a donor site (e.g., part of the palatal mucosa) is transplanted to a recipient site to increase the width of the attached gingiva or to cover a root with gingival recession. A gingival flap is placed firmly in the recipient site to avoid the formation of dead space and is then sutured to the surrounding mucosa. Currently, the main procedure for exposed donor sites is covering them with a periodontal pack or artificial dermis.

On the other hand, a CO_2_ laser can be used in this treatment to form artificial scabs on the surface of blood accumulated at the donor site. These scabs, like a periodontal pack or artificial dermis, are likely to play a role similar to that of a wound dressing (see Section 4.1). A CO_2_ laser can also be used after suturing for soldering wound edges in the spaces between suture points [40,41]; the treatment is completed after performing PBMT several times (Figure 2j–l and Figure 7). We use a scalpel, but not a CO_2_ laser, when harvesting a gingival flap to avoid a risk of protein coagulation by heating due to laser irradiation at the margins of the flap edges.

Studies of CO_2_ lasers in this treatment are very limited. In one study, gingival flaps of good quality without thermal protein denaturation were harvested, and high root coverage and significant gains in keratinized gingiva were achieved [57]. In another study, recession of the gingival flap 1 year after grafting was significantly decreased by postgraft irradiation of the recipient site [58]. However, basic research investigating whether CO_2_ lasers promote the healing of mucosal epithelial wounds and whether they influence scar formation at donor and recipient sites has not yet been reported.

PBMT using a laser that penetrates tissue as an adjunct to wound repair markedly accelerated re-epithelization and contributed to wound pain relief [59]. It was noted that PBMT using a Nd:YAG laser increased the expression of TGF-b1 at the donor site to stimulate fibroblasts to close the wound in the early phase of wound healing, but the TGF-b1 expression was decreased in the late phase [60]. As described in Section 4.2, PBMT may work to avoid scar formation as wound healing progresses to the maturation phase.

## 8. Conclusions

The following are important points and effects of using a CO_2_ laser for secondary healing of small parenchymal defects in oral soft tissues:Blood is allowed to accumulate at the exact site of a small parenchymal defect, and the blood surface is coagulated and carbonized by HILT to form an artificial scab.The artificial scab is soldered to the surrounding mucosa so that it will not separate from the surrounding mucosa.Artificial scabs have a space-making effect for the accumulation of blood and effusion necessary for tissue regeneration.Artificial scabs facilitate the accumulation of blood and effusion, thereby preserving a moist environment (a moist wound-healing-like effect).Artificial scabs protect wounds from contamination and infection.PBMT contributes to suppressing scar formation at the wound site and to promoting wound healing.

Taken together, the use of a CO_2_ laser contributes to the suppression of scar formation, leading to favorable esthetic and functional outcomes of wound healing. However, cases where lasers can be used must be chosen based on a good understanding of the laser characteristics, and evidence-based medicine is essential for choosing suitable cases for laser use, as laser treatment without evidence from clinical and basic research may cause wound healing failure.

## Figures and Tables

**Figure 1 diseases-11-00172-f001:**
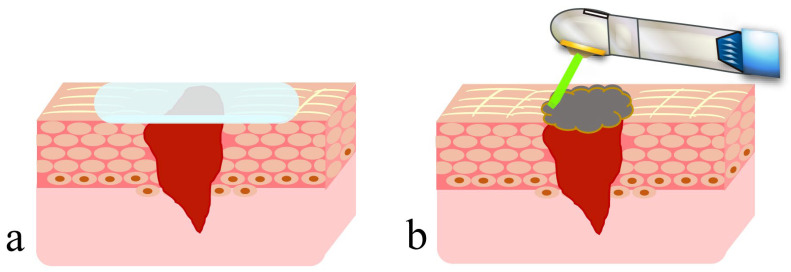
Dressing therapy with wound dressing and wound covering with an artificial scab formed using a CO_2_ laser: (**a**) wound dressing; (**b**) artificial scab.

**Figure 2 diseases-11-00172-f002:**
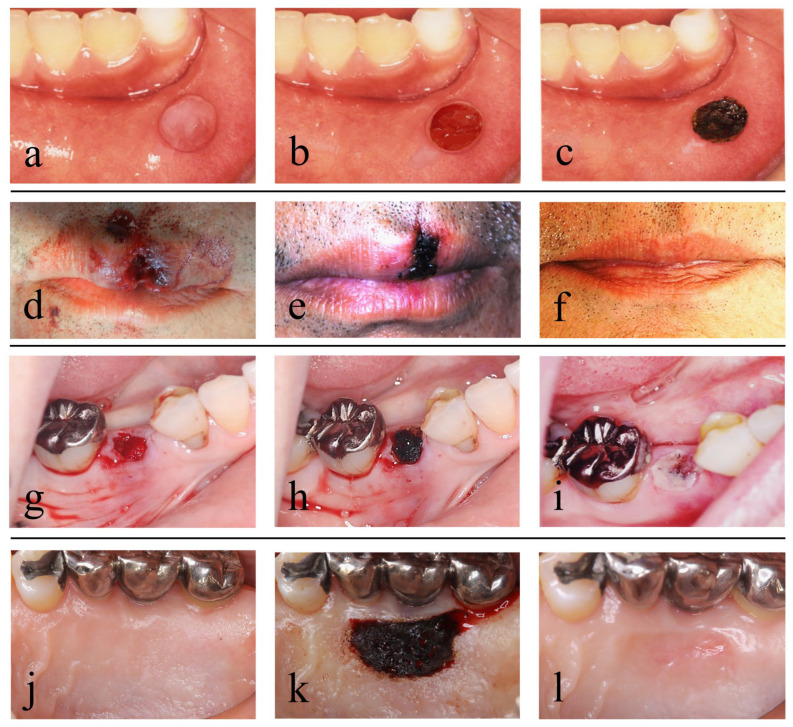
Cases and surgical methods with artificial scabs formed using a CO_2_ laser: (**a**–**c**) removal of a mucocele; (**d**–**f**) treatment of open lip vermillion wounds; (**g**–**i**) treatment after tooth extraction; (**j**–**l**) treatment of the donor site for a free gingival graft; (**a**) preoperative; (**b**) after removal of the mucocele; (**c**) an artificial scab formed on the surface of the resection wound; (**d**) immediately after open lip vermillion wounds; (**e**) artificial scab formed in open lip vermillion wound; (**f**) 6 months after open lip vermillion wound; (**g**) the extraction socket fully filled with blood immediately after tooth extraction; (**h**) an artificial scab formed on the surface of blood filled in the extraction socket; (**i**) day 2 after tooth extraction; (**j**) preoperative; (**k**) an artificial scab formed after the donor site of a gingival flap filled with blood clots; (**l**) 1 month after surgery. Photos by (**d**–**i**) co-author Dr. H. Fukuoka and (**j**–**k**) Dr. Funakoshi. Consent was obtained from the patients in all cases. Reproduced from [29] under Creative Commons CC BY-NC 4.0.

**Figure 3 diseases-11-00172-f003:**
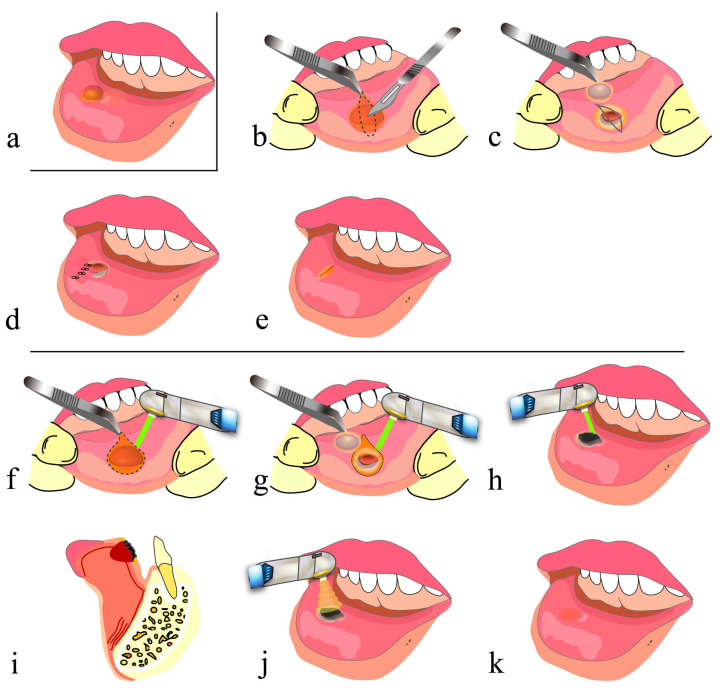
Removal of mucocele using a CO_2_ laser: (**a**) before removal of mucocele; (**b**–**e**) conventional surgical method; (**f**–**k**) surgical method using CO_2_ laser; (**b**) designing a spindle incision line over the mucocele along a lip wrinkle and making a submucosal incision; (**c**) removing a mucocele without damaging it; (**d**) hemostatic suturing; (**e**) high likelihood of formation of a linear scar and a mucosal concavity along the sutured wound edges; (**f**) making an incision in the shape of the mucocele using a CO_2_ laser; (**g**) removing a mucocele as in (**c**); (**h**) letting blood accumulate in the shape of the resection wound, and forming an artificial scab on the surface; (**i**) sagittal cross section of (**h**); (**j**) performing PBMT; (**k**) high likelihood of healing without scar formation.

**Figure 4 diseases-11-00172-f004:**
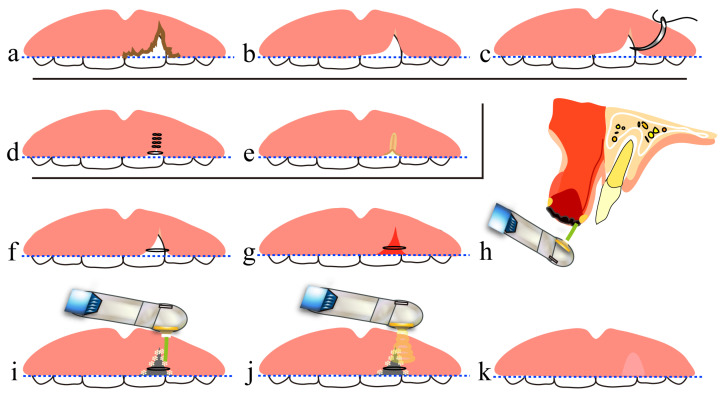
Treatment of open lip vermillion wounds using a CO_2_ laser: (**a**–**c**) procedures common to both hemostatic suturing and treatment with a CO_2_ laser; (**d**,**e**) treatment with hemostatic suturing only; (**f**–**k**) treatment using CO_2_ laser; (**a**) immediately after injury; (**b**) after debridement; (**c**) before hemostatic suturing; (**d**) suturing with prioritization of wound closure but without consideration of esthetic outcomes of the lip; (**e**) an interrupted inferior vermillion border of the upper lip, with a high possibility of healing with a linear scar; (**f**) minimum suturing to make an uninterrupted inferior vermillion border of the upper lip; (**g**) letting blood accumulate in the shape of a parenchymal defect in the tissue; (**h**) sagittal cross section of (**g**); (**i**) forming an artificial scab on the blood at the wound surface; (**j**) performing PBMT; (**k**) high likelihood of healing without mucosal scars. A dotted line indicates the inferior vermillion border of the lip.

**Figure 5 diseases-11-00172-f005:**
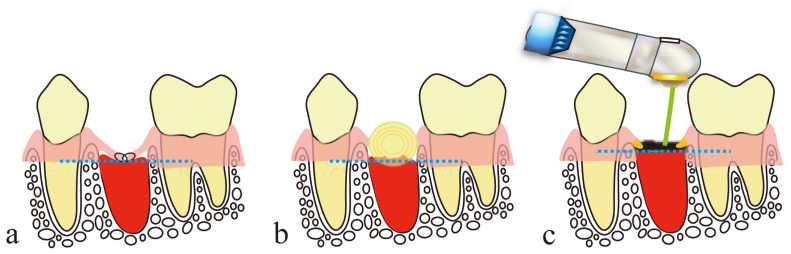
Differences in the amount of blood accumulated in the tooth extraction socket due to different hemostasis methods: (**a**) suturing of the extraction wound; (**b**) compression hemostasis with a cotton roll; (**c**) blood coagulation by artificial scab using a CO_2_ laser. The broken line indicates the position of the blood surface layer within the tooth extraction socket.

**Figure 6 diseases-11-00172-f006:**
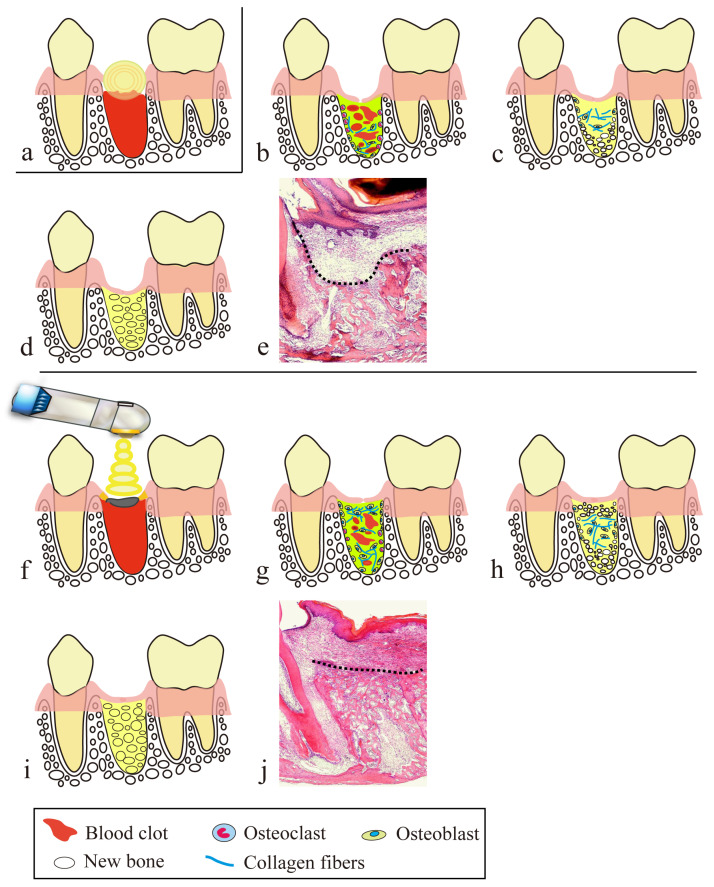
Comparison of tooth extraction wound healing between compression hemostasis and artificial scab formation with a CO_2_ laser: (**a**–**e**) the wound healing process after compression hemostasis with a cotton roll; (**f**–**j**) wound healing process with an artificial scab; (**a**,**f**) coagulation stage; (**b**,**g**) granulation tissue stage; (**c**,**h**) temporary bone stage; (**d**,**i**) healing stage; (**e**,**j**) histopathological image of a tooth extraction wound 7 days after tooth extraction in a rat (corresponds to the temporary bone stage in humans); (**a**) the amount of blood in the extraction socket is decreased due to blood absorption by dental cotton rolls; (**b**) the space-making effect of bone regeneration is decreased due to extension and invagination of mucosal epithelium surrounding the extraction wound into the extraction socket and depression of the extraction wound with contraction of the epithelial scar; (**c**) new bone is regenerated from the bottom of the extraction socket; (**d**) alveolar bone height is decreased and a dish-like concavity forms in the mucosa of the extraction wound; (**e**) histopathological image of a rat tooth extraction socket at the temporary bone stage after compression hemostasis; (**f**) formation of an artificial scab lets blood accumulate to the height of the mucosa around the extraction wound, followed by PBMT; (**g**) the space-making effect for bone regeneration by the artificial scab prevents extension and invagination of mucosal epithelium around the extraction wound into the extraction socket, in combination with suppression of contraction of the mucosal epithelial scar by PBMT. Also, due to the healing-promoting effect of PBMT, the appearance of cells involved in bone regeneration in the shallow layer of the extraction socket and the production of collagen fibers were promoted, followed by the migration of osteoblasts on those collagen fibers; (**h**) new bone forms with a cross-linking pattern in the shallow and middle layers of the extraction socket, in addition to bone regeneration from the bottom of the extraction socket; (**i**) alveolar crest height is preserved and there is no concavity of the mucosal epithelium, due to the newly generated bone with a cross-linking pattern that serves as a bone lining under the mucosal epithelium of the extraction wound; (**j**) histopathological image of a rat tooth extraction socket in temporary bone stage after covering with artificial scab. The broken line in (**e**) and (**j**) indicates the height of the alveolar bone. Reproduced from [26,27] under Creative Commons CC BY-NC 4.0.

**Figure 7 diseases-11-00172-f007:**
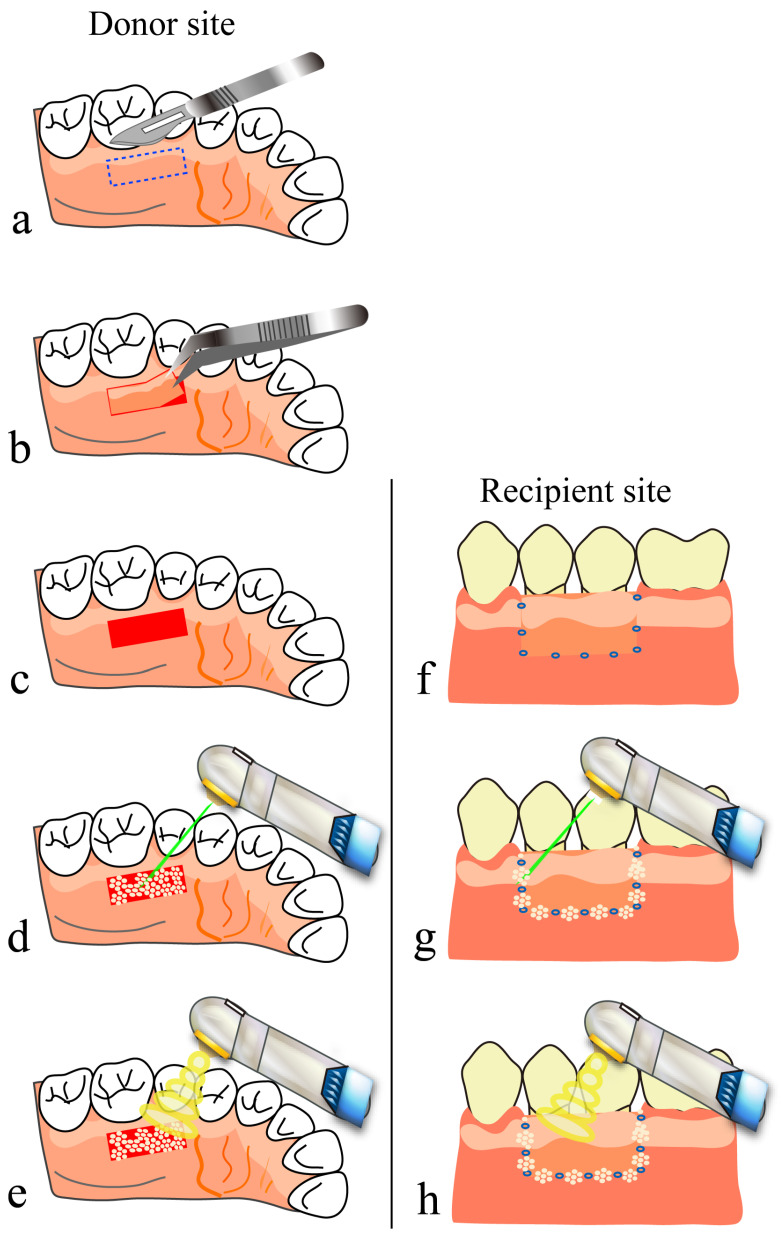
Free gingival graft using a CO_2_ laser: (**a**–**e**) donor site; (**f**–**h**) recipient site; (**a**) determine the area for a gingival flap; (**b**) harvest the gingival flap; (**c**) after harvesting the flap, let blood accumulate on the tissue surface of the donor site to the height of the surrounding mucosa; (**d**) form artificial scabs; (**e**) perform PBMT; (**f**) place the gingival flap firmly in the recipient site to avoid formation of dead space between them, and then suture the flap to the surrounding mucosa; (**g**) solder the wound edges in the spaces between suture points with HILT; (**h**) perform PBMT.

## Data Availability

Data are contained within the article.

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
