# Peer review of "CO2 Laser for Esthetic Healing of Injuries and Surgical Wounds with Small Parenchymal Defects in Oral Soft Tissues"

_diseases, 2023, doi:10.3390/diseases11040172_

Round 1
Reviewer 1 Report
Comments and Suggestions for Authors
The scientific paper “Use of a CO2 Laser for Artificial Scab Formation by High-intensity Laser Therapy and Photobiomodulation Therapy to Suppress Tissue Scarring Leading to Esthetic Healing of Injuries and Surgical Wounds with Small Parenchymal Defects in Oral Soft Tissues” aimed to describe basic research and clinical studies of aesthetic healing using CO2 laser for both the formation of artificial crusts by high-intensity laser therapy and PBMT in the treatment of injuries and surgical wounds with small parenchymal defects in oral soft tissues. I can make the following considerations:
1) The title of the manuscript is very long. Please reduce;
2) The abstract does not contain all the necessary elements. It ends with the objective. Please enter results and conclusion.
3) The manuscript is well written, with detailed and high-quality figures.
Comments on the Quality of English Language
Minor editing
Reviewer 2 Report
Comments and Suggestions for Authors
The submitted manuscript discusses recent studies on using CO2 laser irradiation to repair and regenerate scar tissue in the context of oral mucosa. While it offers valuable information, there are a few areas that can be improved:
[Point 1] Wound healing is a complex process with multiple stages, one of which involves scab formation. The authors correctly emphasize the significance of "artificial scabs" created through mucosal tissue carbonization by CO2 lasers. However, the manuscript also underscores the role of photobiomodulation in suppressing TGF-β1. This presents a nuanced contradiction because, typically, TGF-β1 strongly stimulates scab formation in the early phases of wound healing. To address this potential "conflict" surrounding TGF-β1 levels, it's crucial to provide a reasonable explanation for the dual role of TGF-β1 in wound healing, taking into account its importance in "artificial scabs" and how photobiomodulation influences TGF-β1 in this specific context.
[Point 2] For each of the three TGF-β isoforms, specific roles in wound healing have been described. TGF-β1 is a potent regulator of inflammatory responses and is usually upregulated in the early phases of wound healing. TGF-β1 and TGF-β2 are both involved in recruiting fibroblasts and immune cells from the circulation and the wound edges into the wounded area, thus promoting the formation of granulation tissue and collagen synthesis. However, excessive or chronically elevated TGF-β1 may promote scar formation and fibrosis in adults. In contrast, TGF-β3 may reduce scarring in adults and even promote scarless healing in the fetus.
The submitted manuscript focuses only on the role of photobiomodulation in suppressing TGF-β1 but does not mention the other two subtypes, TGF-β2 and TGF-β3. This aspect should also be mentioned briefly for a more comprehensive understanding of the role of photobiomodulation in dental wound healing.
Reviewer 3 Report
Comments and Suggestions for Authors
It is my great pleasure to have an opportunity to review this interesting article. The authors reviewed the useful indication of CO2 LASER for small defects in oral soft tissue. This review article is well-written, informative, and exciting. It is almost suitable for publication. I have only one minor comment. In Section 4.2, the mechanism of the inhibitory effect of PBMT on TGF-beta signaling should be described in more detail because TGF-beta signaling is known main signaling pathway for scarring.
